# Shrinking to bird size with dinosaur-level cancer defences: Evolution of cancer suppression over macroevolutionary time

**E. Yagmur Erten** [iD] [1,2,3,4*], **Marc Tollis** [iD] [2,5], **Hanna Kokko** [1,6,7]

1 Department of Evolutionary Biology and Environmental Studies, University of Zurich, Zurich, Switzerland, 2 Arizona Cancer Evolution Center, Arizona State University, Tempe, Arizona, United States of America, 3 Department of Biological and Environmental Science, University of Jyväskylä, Jyvaskylä, Finland, 4 Groningen Institute for Evolutionary Life Sciences, University of Groningen, Groningen, The Netherlands, 5 School of Informatics, Computing, and Cyber Systems, Northern Arizona University, Flagstaff, Arizona, United States of America, 6 Institute of Organismic and Molecular Evolution, University of Mainz, Mainz, Germany, 7 Institute for Quantitative and Computational Biosciences, University of Mainz, Mainz, Germany

* e.y.erten@rug.nl

**Data availability statement:** All data, simulation codes, and analysis scripts can be accessed at https://doi.org/10.6084/m9.figshare.25796929.

## Abstract

Ubiquity of cancer across the tree of life yields opportunities to understand variation in cancer defences across species. Peto's paradox, the finding that large-bodied species do not suffer from more cancer despite having more cells at risk of oncogenic mutations compared to small species, can be explained if large size selects for better cancer defences. Since birds live longer than non-flying mammals of equivalent size, and are descendants of moderate-sized dinosaurs, we ask whether ancestral cancer defences are retained if body size shrinks in a lineage. Our model derives selection coefficients and fixation events for changes in cancer defences over macroevolutionary time, based on known relationships between body size, cancer risk, extrinsic mortality, metabolic rate, and effective population size. We show that, if mutation rate is sufficiently high and cancer defences are costly, we expect birds to have lower cancer defences than their dinosaurian ancestors. However, if the evolution of cancer suppression is mutation limited, due to e.g. pleiotropy, birds may have kept excessive dinosaurian cancer defences, possibly explaining their low cancer risk. Counterintuitively, birds can then be 'too robust' for their own good, if excessive cancer suppression requires compromising reproductive rates. Yet, evolutionary innovations such as flight can increase longevity and keep selection for cancer suppression intact in birds, even if flight requires small body size. Retaining dinosaur-level cancer defences can then be adaptive, particularly if the evolution of flight is accompanied by an increase in cancer risk due to metabolic scaling. Overall, our study suggests that studying cancer suppression in birds can reveal alternative mechanisms to those found in mammals, possibly inherited from birds' dinosaurian ancestors.

**Funding:** This work was supported by the university research priority program (URPP) "Evolution in Action" of the University of Zurich (awarded to HK), National Institutes of Health (NIH) grant U54 CA217376 (awarded to HK), and Swiss National Science Foundation (SNSF) grant no. P500PB 203022 (awarded to EYE), the GenEvo graduate school of the University of Mainz, and the Konnevesi Funding Scheme of the University of Jyväskylä. The funders had no role in study design, data collection and analysis, decision to publish, or preparation of the manuscript.

**Competing interests:** The authors have declared that no competing interests exist.

## Author summary

Humans are not the only species suffering from cancer, yet cancer does not impact all life equally. Body size is relevant because maintaining a large number of cells in a tumour-free state necessitates improved cancer defences. Present-day birds appear adept at avoiding cancer, and live longer than similarly sized mammals. Since birds arose from a large-bodied dinosaur lineage rather than having always been small, we ask if evolutionary lags might explain their superior cancer defences. We model the circumstances under which dinosaurian anti-cancer innovations can still persist, making birds more cancer-resistant and longer-lived than mammals. We show that unless the process is mutation-limited, dinosaur-level cancer defences will decay over macroevolutionary time. However, evolutionary lags, when they do occur, can render birds cancer-robust and explain their relatively long lifespans. Eroding defences is made less likely by evolution of flight, which by reducing extrinsic mortality risk can make long lifespans feasible, and by metabolic scaling during shrinkage, if increasing mass-specific metabolic rate translates into an increase in the rate of cancer-causing mutations.

## Introduction

Cancer is likely ubiquitous in nature, having been observed in all but few lineages of multicellular organisms [1,2]. Multicellularity relies on cooperation between the cells of an organism, and mutations that disrupt this cooperation can cause uncontrolled cell proliferation and oncogenesis [1,3]. These mutations can arise during cell divisions, as well as due to errors in DNA damage repair between cell divisions, putting both mitotic and post-mitotic cells at risk of accumulating oncogenic mutations [4,5]. Since larger and longer-lived organisms need more divisions to reach and maintain their adult body size, and have more cells susceptible to DNA damage, the 'all else being equal' expectation is a higher cancer risk if either body size or lifespan increases. Therefore, cancer can become a limiting factor to a further evolutionary increase of either trait [6,7]. Studies in humans [8–10] and dogs [11,12] suggest that larger individuals within a species have an increased cancer incidence. However, the between-species pattern is different: large body size and long lifespan do not combine to predict elevated cancer incidence ([13–15], but see [16–18]). This lack of support for a theoretically clear *a priori* expectation is known as 'Peto's paradox' [19,20].

Current comparative oncology efforts to understand cancer across the tree of life are based on the insight that species differ in past selection to improve cancer suppression mechanisms. Such selection may be stronger in large-bodied and/or long-lived species [21,22], and these species make fitting targets for understanding the molecular bases of cancer defences [23–28]. For instance, higher rates of p53-mediated apoptosis in response to DNA damage in elephant cells and multiple copies of the tumor suppressor gene TP53 in the elephant genome [13]. Evidence also comes from smaller but unusually long-lived organisms: the long and relatively cancer-free life of naked mole rats (*Heterocephalus glaber*) has been attributed to their skin cells showing a high degree of contact inhibition which provides cancer resistance— important in a species in which reproductives, so-called queens, have exceptional longevity for their body size [29]. Interestingly, non-reproductive naked mole rats, so-called workers, also live unusually long for their body size when in captivity, with very rare occurences of tumours. It is probably relevant that reproductive individuals derive from workers, and long life may thus improve the chances that a worker switches to being a breeder [30].

Lifespan correlates strongly with body size, with larger animals typically living longer [31–33]. Yet, there are tantalizing differences between taxa: for their (usually small) body size, birds are relatively long lived [34–37], and compared with mammals, they enjoy a delayed onset [38,39] as well as a slower rate of senescence [40,41]. Ages beyond 60 years old can be reached by some species. Flight has been argued to be a key innovation [34] that allows birds to experience lowered extrinsic mortality (e.g. from predation or starvation), a pattern repeated in flying mammals [35,37,42] which, too, live long for their body size. A 'slow' life history in a flying organism can only reap the benefit of a potentially long life if its body is robust enough to escape senescence (for some time) too, creating the so-called Williams prediction [43]: under many realistic ecological conditions, low mortality selects for delayed senescence [44].

One subtask within senescence avoidance is to avoid succumbing to age-related diseases such as cancer, as theory and empirical data suggest cancer incidence increases with age [45–47]. Current data availability renders between-taxa comparisons difficult and may result in under- or over-representation of neoplasia in some species unless life history traits such as longevity are also considered. Nevertheless, the relatively low cancer incidence of birds compared with mammals and reptiles [16,17,48,49] is intriguing. A recent study has shown that cancer prevalence in birds was not correlated with body size and longevity (in line with Peto's paradox), but increased with clutch size [50]. Yet, as a whole, avian cancer suppression mechanisms remain unexplored. Comparative studies show that bird cells are more stress resistant than rodent cells [51,52], but the interpretation is challenging: as the elephant example shows, more (rather than less) cell-level death might improve longevity at the organismal level.

In addition to their apparent cancer-robustness, birds are also intriguing due to their evolutionary past. All extant birds belong to a lineage that shrunk from large theropod ancestors through a process of continuous miniaturisation, accompanied by innovations that eventually led to flight [53,54]. Based on the prevalence of tumours across the tree of life [1], it is parsimonious to assume cancer was also a problem affecting dinosaurs (for direct evidence see [55–58]). The ancestors of birds were estimated to be ∼163 kg around 198 million years ago [54], before miniaturisation began, which implies that the appropriate cancer defences must have been in place (analogously to the argument that extant whale life histories would be impossible, based on colorectal cancer alone, if whales did not have better cancer defences than humans; [22]). Yet, an *a priori* beneficial trait—here, any form of cancer defence—may lose importance when body size shrinks [6].

Could modern bird life histories be shaped by having inherited pre-existing anti-cancer adaptations from their Mesozoic ancestors? This would mean they may be 'too robustly built' for the needs of their current life history, but simultaneously free to explore, when ecological conditions permit, the slower end of the pace-of-life continuum. Birds inherited dinosaur genes from very long ago, raising questions about the timescales over which evolutionary lags can operate. In the case of cancer-related adaptations, evolutionary lags occur if gains or losses in cancer defences happen at a slower rate than evolutionary changes in body size and life history [59]. We thus examine under what conditions evolutionary lags in cancer-related adaptations are expected to influence extant birds' life history and relatively low cancer incidence.

Specifically, we ask: 1) how quickly will defences against cancer adapt when a lineage changes in terms of body size, and under what conditions does this result in evolutionary lags; 2) how do innovations reducing extrinsic mortality (evolution of flight) affect evolutionary lags in cancer-related adaptations?; 3) can an increase in the rate of oncogenesis due to metabolic scaling diminish evolutionary lags? To answer these questions, we model a mutation-selection process over macroevolutionary time scales, spanning several magnitudes

of mutation rates. We assume allometric relationships that relate body size to effective population size and extrinsic mortality, which in turn impact the probability of fixation of mutations that alter the expected time that an individual body can resist cancer.

Our results show that under realistic mutation rates, dinosaurian cancer adaptations decay and birds evolve lower cancer suppression than their ancestors. However, if the evolution of cancer defences is constrained (e.g. by mutation rate and/or pleiotropic effects), cancer adaptations can lag behind the body size change in birds. This may result in a longer lifespan and a lower cancer risk, but could also reduce the reproductive success (assuming trade-offs exist between these life history components, i.e. cancer defences are costly; [60]). On the other hand, increase in mass-specific metabolic rate and innovations that lengthen the average lifespan can select for high cancer suppression, thereby diminishing the effect of these evolutionary lags.

## Methods

### Model overview

We simulate how cancer defence innovations arise and/or decay in a population via mutations (innovation-decay process, detailed in section *Macroevolutionary simulation*) for 240 million years. We define the relationship between cancer defence and lifespan following the derivation of [6]. We follow [60] in assuming that cancer defences represent a shift towards a slower life history: strong defence yields a (probabilistically) longer lifespan but at a cost of reproductive success per unit time while the organism was alive (implemented as described in section *Deriving the selection coefficient of a mutation*). We assume extrinsic mortality rate scales with body size following the allometric relationships from the literature ([61]; see section *Life history assumptions* below). In different variants of our model, we model metabolic scaling during miniaturisation and associated change in the rate of oncogenesis ([62]; see section *Life history assumptions* below), and/or implement flight as an innovation that can reduce the extrinsic mortality rate (described in section *Modelling body size change and flight*).

We vary the overall mutation rate across scenarios while keeping the distribution of increases and decreases in cancer defences unchanged. At each time point, we calculate the selection coefficient for each potential mutation by examining how it changes lifetime fitness when the life history of the carrier otherwise follows that established in the current population (body size, extrinsic mortality, presence or absence of flight), but with altered lifespan and reproductive success as these are impacted by the mutation. The fixation probability of cancer-related mutations depends on this selection coefficient, but also on effective population size, which in turn depends on body size based on known ecological relationships. We present an overview of the model in Fig 1.

### Modelling body size change and flight

To disentangle the effect of diminishing body sizes (*M*, see Table 1 for notation) from other evolutionary changes over time, we compare results obtained with 1) a realistic timeline of body sizes shrinking with time ('miniaturisation' scenario) with two hypothetical "control" scenarios that keep body sizes constant either at the 2) ancestral size (the 'dinosaur-like' constant large flightless lineage, *M* = 220.7 kg), 3) final size of the miniaturised lineage (the 'mammal-like' constant small flightless lineage, *M* = 0.8 kg) for 240 million years.

We do not model selection that led to the miniaturisation of birds explicitly, instead we approximately follow the phenomenologically documented pattern [54] from theropod ancestors to a typical extant bird (Fig 1A). We use the dates reported in the text on [54] with the

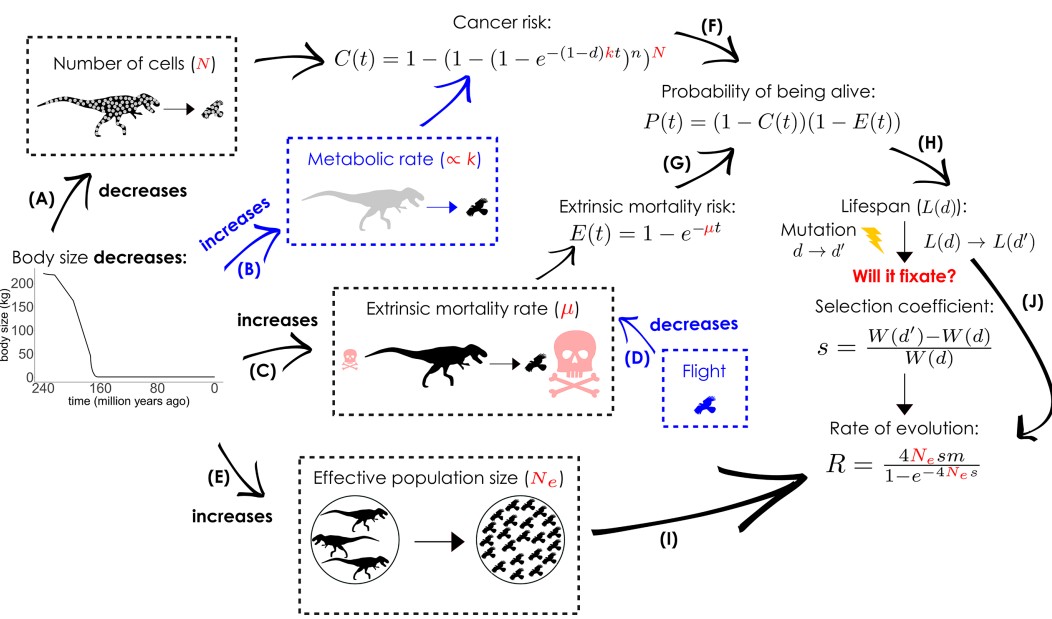

**Fig 1. Model overview, with model variants indicated in blue.** Body size, *M* (model notations in Table 1), gradually decreases from 240 Mya to 163 Mya, approximated from [54], stabilizing thereafter. As the body size shrinks, A) the number of cells decreases, lowering the age-specific cumulative cancer risk; B) in a variant of the model, the mass-specific metabolic rate increases, thereby increasing cumulative cancer risk, assuming a correlation between the metabolic rate and the rate of oncogenesis; C) the ecology changes to increase the extrinsic mortality rate; D) but with an optional permanent mortality reduction when the innovation of flight happens; E) population density increases, resulting in a larger effective population size $N_e$. An individual can reproduce as long as it is alive, the age-specific probability of which is impacted by probability of death (by time *t*) by F) cancer $C(t)$ and by G) extrinsic mortality $E(t)$, then H) used to calculate lifespan $L(d)$. Each mutation is associated with a selection coefficient based on [6], I) which is used together with $N_e$ to arrive at a rate of this mutation fixing. Mutation rate *m* in our simulations is per genome per generation, therefore, J) to obtain the rate of at which mutations in cancer defences will arise and fixate (*R*) we use the population's currently evolved lifespan as a proxy for the generation time. Bird and tyrannosaur silhouttes are adapted from Phylopic and Wikimedia Commons, respectively.

femur lengths shown in their Fig 1 (converted to body size in kg with the formula in their supplementary material). With PlotDigitizer (http://plotdigitizer.sourceforge.net/), we also extract femur length and dates reported in Fig 2B (dataset 1) of [54]. We report all the dates and body sizes in the S1 Data. Hence, in our model, the bird lineage starts at 220.7 kg 240 million years ago (Mya), and reaches the small body size of 0.8 kg at 163 Mya. We assume linear body size shrinkage between any two consecutive dates with body size information. Once miniaturised, we assume the bird lineage not to change in size for 163 million years. We thus ignore extant bird body size variation.

We derive results for our miniaturisation scenario both with and without flight acquisition, to examine how a one-off innovation that reduces extrinsic mortality can affect evolutionary lags in life histories. If lower extrinsic mortality prolongs the expected lifespan, then flight might temporarily increase the relative importance of cancer as a cause of death, prompting selection to increase defences. However, this effect may be absent if flight acquisition occurred when cancer defences were lagging behind, being too strong for the current needs in a miniaturizing lineage. This model variant operates like the baseline miniaturisation scenario, except that from the point onwards where flight innovations occur, the extrinsic mortality is multiplied by a coefficient *r* < 1 and is thus reduced to a fraction of what the model otherwise assigns to an animal of a specific body size. The associations between size and mortality still

**Table 1. Model notation.**

| Symbol | Meaning |
|---|---|
| $M$ | Body size (in kg) |
| $N$ | Number of cells |
| $\mu$ | Extrinsic mortality rate |
| $N_e$ | Effective population size |
| $t, T$ | time (age of the organism, and year in macroevolutionary time, respectively) |
| $C(t), E(t)$ | Probability of death by time $t$ due to cancer and extrinsic causes, respectively |
| $k$ | Rate of oncogenesis |
| $n$ | Number of oncogenic steps until cancer |
| $d$ | Level of defences against cancer |
| $P(t)$ | Probability of being alive at age $t$ |
| $L(d), W(d)$ | Lifespan and fitness (lifetime reproductive success), respectively, for a given level of cancer defence |
| $c$ | Cost parameter determining the strength of trade-off between cancer defences and fitness |
| $R$ | Rate of evolution; the rate at which mutations that change cancer defences arise and become fixed in the population |
| $m$ | Mutation rate per genome per generation |
| $\tau$ | Time (in generations) until a mutation fixes in simulations |
| $r$ | Extrinsic mortality reduction coefficient |
| $\kappa$ | Oncogenic rate scaling coefficient |

operate after flight acquisition. [54] dates the bird node to ∼163 Mya and recent studies suggest powered flight potential in *Archaeopteryx* [63,64], as well as other forms of wing-assisted movement in various early ancestors of birds [64]. Therefore, we chose three time points (180, 170 and 160 Mya) to simulate the initiation of flight.

## Life history assumptions

Various allometric relations exist between body size and physiological and ecological traits [31]. First, shrinking translates into a lower number of cells $N$, with cell number scaling linearly with body size (Fig 1A). We calculate $N$ that corresponds to a given body size $M$ using estimates made for humans [65], $N = M/70 \times 3.72 \times 10^{13}$, assuming an average human weight of 70 kg. Second, smaller organisms have a higher death rate, mainly due to extrinsic causes such as predation (Fig 1C; [31]). With the relationship between body size and mortality from [61], we calculate the extrinsic mortality as $\mu = e^{-1.8} M^{-0.21}$. In the model variant with flight innovation (Fig 1D), we multiply the extrinsic mortality by a coefficient $r < 1$ for those time points where flight has already evolved, as described above.

Third, because the fate of mutations with positive or negative selection coefficients depends on effective population size ($N_e$, details described below in section *Macroevolutionary simulation*), we use known ecological relationships between body size and population density [66,67]. We assume that miniaturisation allows a species to reach higher densities, while the species range and the relationship between census population size and effective population size are left unchanged (such that we assume no dramatic changes in the mating system and other factors that may impact the relationship between census population size and $N_e$). Thus, effective population size $N_e$ is assumed to scale with body size with the same inherent non-linearity (same value of exponent) as the relationship between body size and density reported by [67]. This results in a larger $N_e$ as the body size shrinks (Fig 1E). For our main examples, we use $N_e = 10^{4-0.49(\log_{10} M + 3) + 1.96}$, which uses the reported densities for $M$ (converted to grams) and derives $N_e$ by making assumptions about the total range of a species as well as a

linear relationship between total (census) population size and $N_e$. The coefficient $10^4$ corresponds to an area of $10^4$ km$^2$ if $N_e$ equals census population size, or to a larger area if $N_e$ falls below census population size. Since our choice of species range area is somewhat arbitrary, the relevant check is whether changing the coefficient $10^4$ affects the main results of our study qualitatively; it does not (Fig A in S1 Text).

Finally, in a variant of our model, we assume that mass-specific metabolic rate increases with the body size shrinkage, resulting in an increase in cumulative cancer risk, assuming rate of oncogenesis is proportional to the mass-specific metabolic rate (Fig 1B). The dinosaur lineage ancestral to birds had evolved endothermy prior to miniaturisation and exhibited high metabolic rates, comparable to extant birds [68]. Therefore, we make the simplifying assumption that there were no major thermoregulatory innovations during the minisaturisation process and assume known allometric relationships between body size and metabolic rate [69]. To implement this, following [62], we calculate the rate of oncogenic steps as $k = \kappa M^{-0.3}$. In this model variant, the coefficient $\kappa$ scales $k$ such that the rate of oncogenic steps at the beginning of miniaturisation is equal to the $k$ used in the simulations without metabolic scaling, to facilitate comparisons between different model variants.

## Deriving the selection coefficient of a mutation

We track the phenotype 'level of cancer defence', $d$, over time, together with its pleiotropic effects on lifespan and reproduction. Mutations change $d$ either upwards or downwards, and either type of change may be selected for or against: e.g. a defence-improving mutation may be selected against if maintaining high defences is costly in a situation where cancer is *a priori* rare, such as in very small-bodied organisms. The life history effects of each mutation consist of increased or reduced expected lifespan, as well as the fecundity cost associated with expressing a specific level of defence. The lifespan effects are modulated by extrinsic mortality: since cancer and extrinsic mortality are two 'competing' ways to die, organisms with low extrinsic mortality can reap benefits of high cancer suppression for a longer time than those with high extrinsic mortality, all else being equal [6].

To compute the lifespan for any level of $d$, we first follow the tradition of modeling cancer as a multi-step mutational process [20,45,70]. We implement a variation of the multistage model [60] that accounts for reproductive trade-offs and calculate the probability that an individual with $N$ cells, alive at time $t$, has cancer by the time it has lived for $t$ units of time, as:

$$C(t) = 1 - \left(1 - \left(1 - e^{-(1-d)kt}\right)^n\right)^N \tag{1}$$

where $d$ is the current level of cancer defences, $k$ determines the rate at which oncogenic steps occur, and $n$ is the number of oncogenic steps that a cell lineage has to complete for the organism to succumb to cancer. As in [60], the probability that a single oncogenic step has happened by age $t$ is $1-e^{-(1-d)kt}$. Here, stronger cancer defences, higher $d$, reduces the effective rate of oncogenesis. With probability $(1-e^{-(1-d)kt})^n$, all $n$ steps will have happened by age $t$, whereas the probability that none of them has happened is given by the complementary probability, $1-(1-e^{-(1-d)kt})^n$. With this, we can calculate the probability that all cells are healthy as $(1-(1-e^{-(1-d)kt})^n)^N$, with its complementary probability being $C(t)$, as above.

The age-dependent increase in cancer risk occurs faster when the number of cells $N$ is large, thus defence $d$ has stronger effects on lifespan in large organisms than in small ones. High extrinsic mortality, by virtue of associating with small body size, has a similar effect, with $E(t) = 1 - e^{-\mu t}$ expressing the probability that an individual is dead by the age $t$. The high

mortality $\mu$ of small-bodied organisms is optionally reduced (to some extent) from the time point where flying evolves (as described in *Modelling body size change and flight* above).

An individual is alive at age $t$, i.e. has succumbed to neither cancer nor extrinsic mortality (Fig 1F and 1G), with probability $P(t) = (1 - E(t))(1 - C(t))$. Integrating this expression, as described in [60], we can express the expected lifespan as a function of defences against cancer, $L(d)$ (Fig 1H):

$$L(d) = \int_{P=0}^{1} 1 - E(t(P,d))\,dP \qquad (2)$$

where

$$t(P,d) = -\frac{ln\left(1 - \left(1 - (1 - P)^{1/N}\right)^{1/n}\right)}{(1 - d)k} \qquad (3)$$

following [60].

Adaptations to reduce cancer risk can be costly and affect the reproductive output of the organism. We follow [60] and model the relationship between fitness (lifetime reproductive success) and cancer defences as $W(d) = L(d)\left(1 - d^{(1/c)}\right)$, where $c$ is a cost parameter that determines the strength of the trade-off between cancer defences and reproductive success per unit time.

For any current level of defence $d$, we consider an identical mutational distribution of potential changes $\delta$: $\delta$ obeys a generalized extreme value random distribution with parameters $k = -0.75$, $\sigma = 1$, $\mu = -1$, which yields values of $\delta$ between $-1$ and $1$. Positive mutations ($\delta > 0$) modify $d$ toward the maximum value $1$ such that the new value $d' = d + \delta(1 - d)$, negative mutations ($\delta < 0$) push $d$ towards zero with $d' = d(1 + \delta)$, yielding $d' = 0$ if $\delta = -1$. These rules combine the plausible assumptions that (i) deleterious mutations (with respect to the level of $d$) are more common than mutations that improve defences, (ii) large increases in cancer defences are rare (for an alternative distribution, see Fig B in S1 Text), and (iii) the defences as a whole remain between 0 and 1 for the cancer risk equation (1) to make sense. We then calculate the selection coefficient for a given change $\delta$ in cancer defences as: $s = \frac{W(d') - W(d)}{W(d)}$ following [6].

## Macroevolutionary simulation

We divide the macroevolutionary time into 4000 bins of 60000 years each, assuming negligible body size change within a bin. The number of time bins does not change our results qualitatively (Fig C in S1 Text). The number of generations that occupy one bin is not predefined, as it changes over time due to changes in mean lifespan. Each step in the simulation consists of mutations competing to fix (at rates that depend on the selection coefficient associated with each potential mutation, and the rate at which mutations that diminish or enhance cancer defences occur), determining the next one that fixes, and shifting the macroevolutionary time to the year when the fixing, stochastically, happened.

Each mutation in the mutational distribution associates with a selection coefficient ($s$) as previously derived in section '*Deriving the selection coefficient of a mutation*', with a rate of such mutations occurring determined by the distribution above, and with a consequent rate of fixing $R$ based on [71]: $R = \frac{4N_e s m}{1 - e^{-4N_e s}}$, where $m$ is the mutation rate per generation, and $N_e$ is the effective population size. Since $R$ depends on the selection coefficient, it is not constant across the different mutations that may arise. We therefore discretize the mutation distribution into 1000 categories by considering mutations that form the 0.05th, 1.5th, …, 99.95th percentile

of the mutation distribution, each of them occurring with mutation rate $m/1000$, and having a rate $R_i$ ($i = 1, 2, \ldots, 1000$) separately derived for each of them according to each mutation's $s$. While mutations with a high $R_i$ have a better chance to outcompete others (fix before another did), we model the stochastic nature of this competition with a Gillespie algorithm [72]: the time until a mutation of category $i$ fixes is negatively exponentially distributed with mean generation time $\tau_i = 1/R_i$; we draw these times for all $i$, and the shortest time found, $\tau_{\text{first}}$, determines the fixation event that actually happens (ahead of any others, which are discarded). Since $\tau_{\text{first}}$ is expressed in the number of generations, we convert this to years by multiplying with the lifespan $L(d)$ as a proxy for generation time (using the value of $d$ before the change in cancer defences) (Fig 1J); $\Delta T = \tau_{\text{first}} \times L(d)$.

The algorithm then updates macroevolutionary time from its previous value to $T + \Delta T$ ($T$ being always expressed in years). If the new time is within the same time bin, the post-mutation cancer defence is set to be the new current defence, and the current lifespan is calculated accordingly. If it is not, we instead keep the defence unchanged, update $T$ to equal the starting year of the next bin and body size to the pre-assigned size that applies within the new bin. In this case, $L(d)$, extrinsic mortality rate $\mu$, and $N_e$ are also updated according to the new body size. The entire procedure is repeated as many times as is needed to cover the time from 240Mya to present.

Each body size and flight scenario is initiated with no evolutionary lag at 240Mya (i.e. we assign the level of cancer defences that maximizes the fitness $W(d)$). Following [73], we use a gradient design for exploring the parameter space; we simulate each parameter combination without replication, and instead cover a large range with finely adjacent values, as this shows general patterns simultaneously with the degree of inherent variation.

## Evolutionary lags

We quantify the evolutionary lags in fitness, lifespan, and cancer defences, to measure potentially maladapted cancer suppression levels when body size and/or flight ability have evolved differences to ancestral conditions. We first calculate the optimal (fitness-maximizing) level of cancer defences $d^*$, denoting the associated fitness as $W_{\max} = W(d^*)$, of the organism for its body size and extrinsic mortality. Similarly, we obtain $L^*$ as the lifespan under optimal level of cancer defence. Then, we define the lag in defences as $\Delta_d = d_{\text{final}} - d^*$, where $d_{\text{final}}$ is the level of cancer defence at the end of the simulations (when $T$=240 million years). Following the same logic, lag in fitness is $\Delta_W = W_{\text{final}}/W(d^*)$, the ratio between the final fitness $W_{\text{final}}$ and the maximum fitness $W(d^*)$, and lag in lifespan is the ratio $\Delta_L = L_{\text{final}}/L(d^*)$ (see Table 2 for summary).

## Results

### Reduced optimal cancer suppression in shrinking lineages

Here, we studied the evolution of cancer suppression over macroevolutionary time in the miniaturising theropod dinosaur lineage that gave rise to extant birds. We asked under which

**Table 2**. Lag definitions.

| Symbol | Meaning |
|---|---|
| $\Delta_d$ | Lag in defences; $d_{\text{final}} - d^*$ |
| $\Delta_L$ | Lag in lifespan; $L_{\text{final}}/L(d^*)$ |
| $\Delta_W$ | Lag in fitness; $W_{\text{final}}/W(d^*)$ |

circumstances the apparent cancer-robustness of birds can be explained by evolutionary lags in cancer defence adaptations and what the consequences of such lags would be. We assumed that the bird lineage underwent a gradual decrease in body size, which shifted two sources of death in opposite directions: extrinsic mortality increased, whereas cancer risk decreased, compared to the birds' dinosaurian ancestors. As a result, the optimal level of cancer suppression reduced in the miniaturised lineages, unless cancer defences had very little cost (Fig 2A).

The optimal level of cancer suppression was higher in lineages that have evolved flight ('flighted' compared to 'flightless'; Fig 2A) or had an increase in mass-specific metabolic rate and an associated increase in the rate of oncogenesis with miniaturisation ('scaling $k$' compared to 'constant k'; Fig 2A). As further discussed below, these factors select for higher cancer suppression either by extending lifespan (flight) or increasing cancer risk ('scaling $k$'). Notably, the combination of flight and scaling $k$ resulted in a level of optimal cancer suppression much closer to that of the 'dinosaur-like' lineage that remained flightless and constantly large-sized over macroevolutionary time ('flighted, scaling $k$' compared to the black line in Fig 2A).

## Low mutation rate can result in phylogenetic inertia in cancer defences

Mutation rate determined whether cancer defences could track their optimal level through substitution processes over macroevolutionary time. Below a certain mutation rate, no substitutions occurred (grey in Fig 2B); we refer to this as the *'mutational boundary'*. Here, the miniaturised lineages kept the ancestral dinosaurian defences (the parts of Fig 2C that correspond to the grey areas in Fig 2B).

Costliness of cancer suppression determined the position of the mutational boundary. The mutational boundary positioned similarly when cancer defences imposed little cost on reproductive success (leftmost part of each subplot in Fig 2B–2C). At this level of cost, the shrinking lineages experienced negligible selection for reducing their cancer suppression since all lineages had similarly high level of optimal defences (Fig 2A). The situation changed in the more realistic scenario where the defences were more costly. Further the lineage was from its optimal level of cancer defences (Fig 2A), stronger the selection was, which pushed the mutational boundary to the lower values of mutation rate (rightmost part of each subplot in Fig 2B–2C).

A closer look at the substitution process over time reveals three distinct mutational dynamics depending on the mutation rate (Fig 2D). Below the mutational boundary, cancer defences did not evolve, even when the ancestral cancer defence was substantially higher than optimal (see Fig 2E for an example). Above the mutational boundary and at high mutation rates, substitutions occurred at a sufficient frequency and kept lineages at a mutation-selection balance. This meant that birds' cancer defences closely tracked the shift in their optimal level of cancer suppression via small mutational steps (see Fig 2I–2J for two examples). Between these two extremes, intermediate mutation rates just above the mutational boundary led to a few substitutions that sometimes occurred via large mutational steps and the resulting cancer defences occasionally over- or undershot the optimal levels (see Fig 2F–2H for some examples).

## Phylogenetic inertia can translate into evolutionary mismatches

Costliness of cancer suppression affected the consequences of keeping ancestral defences. The nearly cost-free expression of strong defences meant that being constrained to express ancestral defences (maintained due to phylogenetic inertia) did not incur large fitness losses for

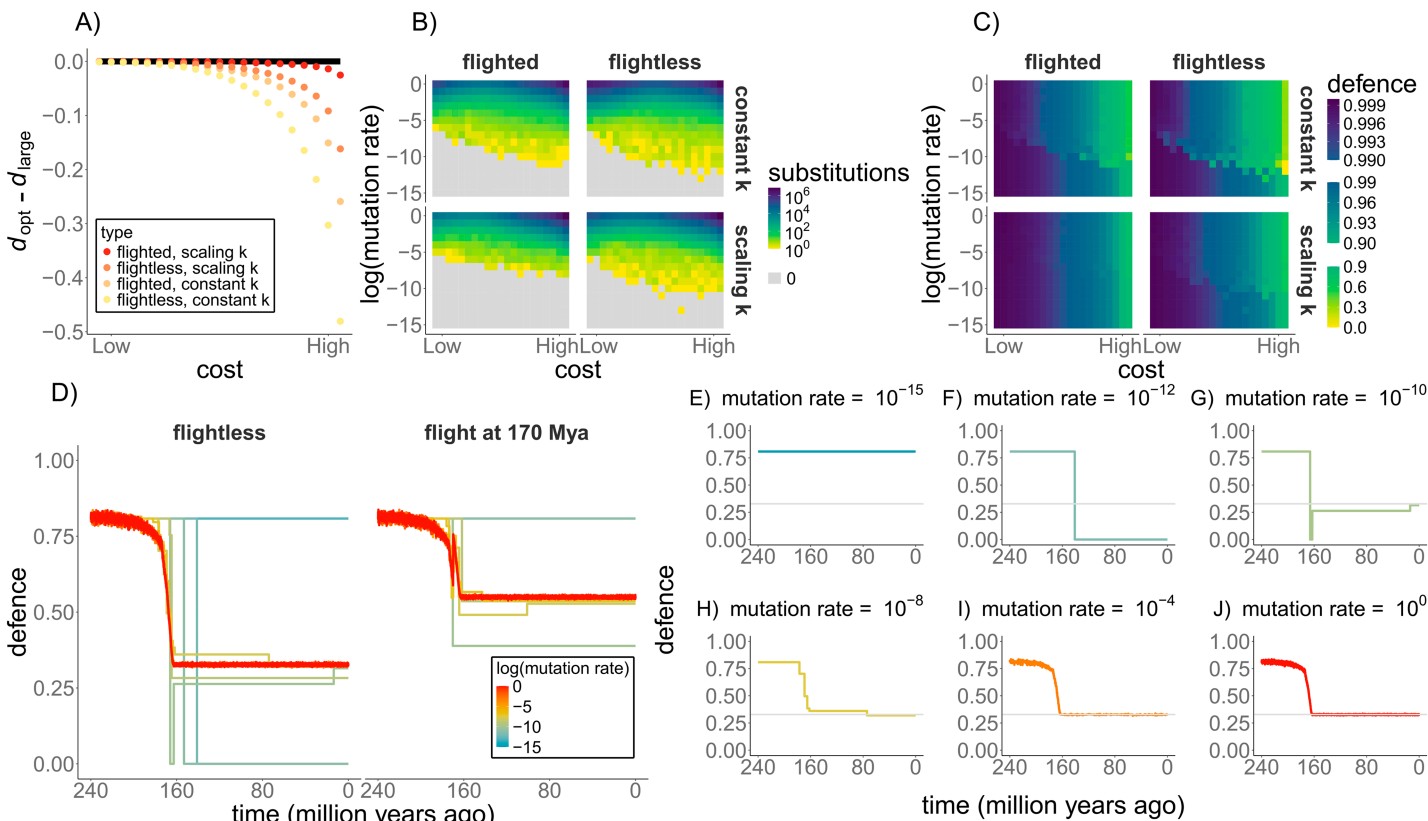

**Fig 2. Evolutionary dynamics of cancer defences depend on the modelling scenario, mutation rate and costliness of cancer defences.** A) The difference between the optimal level of cancer defences of the miniaturised lineage in different scenarios (as indicated by the colours in the legend) and the 'dinosaur-like' constant large flightless lineage (black line), across different cost values (x axis: $\log_{10} c$ ranging between -4.1 and -0.1). The optimal cancer suppression of the constant small flightless lineage is the same as the flightless miniaturised lineage with constant $k$ (yellow), and is therefore not included in this figure. B) The total number of mutations that became fixed by the end of each simulation (grey denotes no fixations), and C) the level of cancer defences at the end of simulations, across different cost values (x-axis as per panel A) and mutation rates (y axis: $\log_{10} m$ ranging between -15 and 0). Smaller panels within B) and C) refer to different scenarios as indicated. D) Cancer defence evolution across macroevolutionariy time for the flightless and flighted lineage with no metabolic scaling and with the cost value $\log_{10} c$ of -0.1. Mutation rate as depicted in the legend, E-J) Cancer defence evolution trajectories for the flightless lineage in D), shown individually. For the panels d-j, y-axis shows the level of cancer defence, x-axis time. Results are reported for the number of oncogenic steps $n = 3$, the rate of oncogenesis $k = 0.0001$ for all scenarios, and extrinsic mortality reduction coefficient $r = 1/3$ for the flighted case.

the miniaturised lineages (leftmost part of each panel in Fig 3A). Here, the lags in lifespan (Fig 3B), and the probability of cancer ending lifespan ($p_{CEL}$, Fig 3C) were also similarly low across all body sizes and scenarios.

When cancer suppression incurred a higher cost (highest at the rightmost part of every panel in Fig 3), maintaining ancestral defences became maladaptive and incurred fitness costs (shown in red in Fig 3A), even as they prolonged lifespan (blue in Fig 3B) and reduced the probability of dying from cancer (Fig 3C). Excessive cancer defences permanently lowered the fitness in the miniaturised lineage (but fitness lags were strongest at moderate rather than high costs; darker red in Fig 3A). This is due to the high extrinsic mortality, particularly in the flightless miniaturised lineage with no metabolic scaling (top right smaller panel in Fig 3A), which made individuals typically not live long enough to enjoy the benefits of high cancer suppression (Fig 3D), while the costs of suppression were paid throughout the lifetime (reduced reproductive success). In other words, most shrunk lineages were too cancer-robust for their own good.

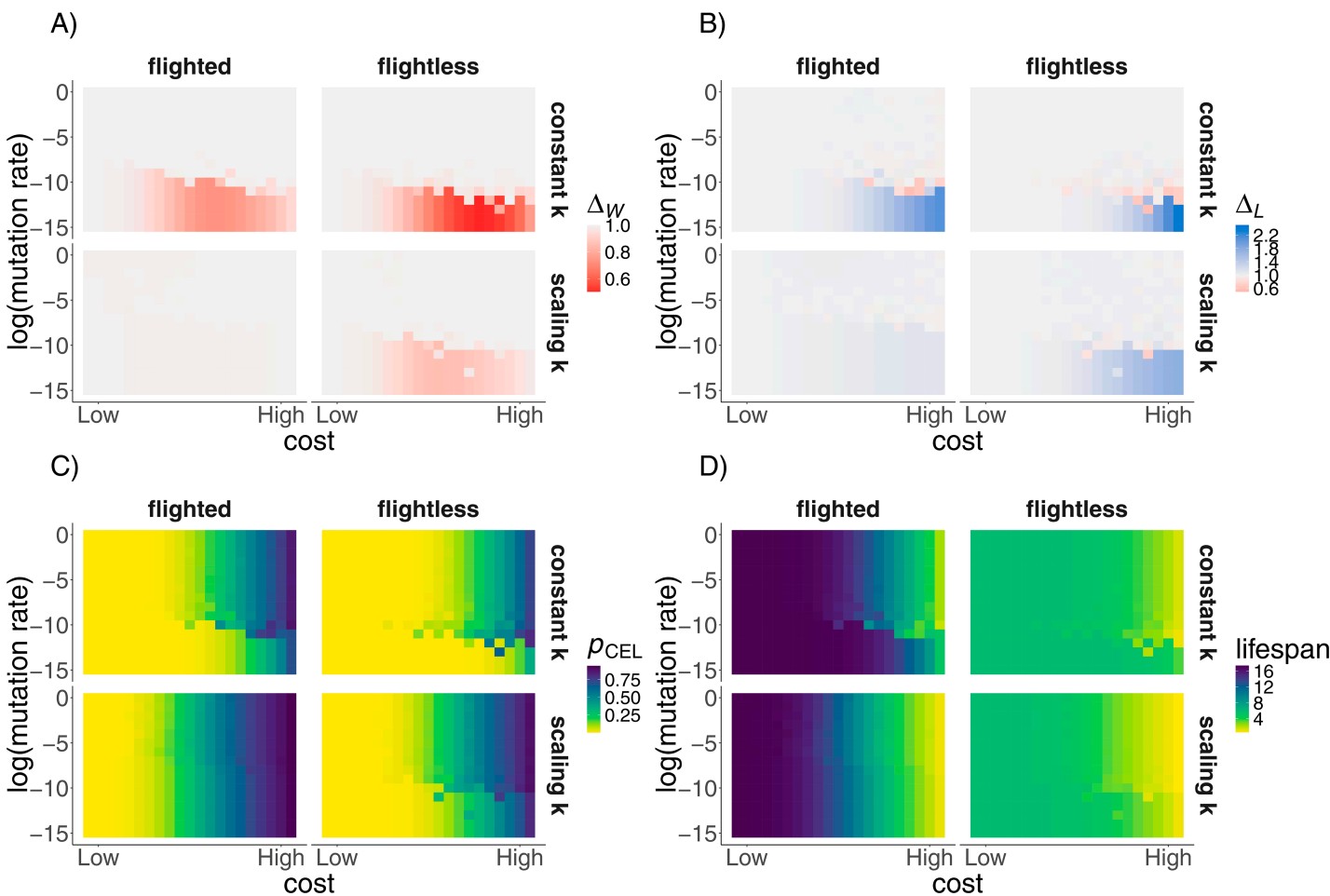

**Fig 3. Phylogenetic inertia in cancer defence evolution affects fitness, lifespan, and cancer mortality.** Evolutionary lags in A) fitness ($\Delta_W$) and B) lifespan ($\Delta_L$), C) probability of dying from cancer ($p_{CEL}$), and D) lifespan (in years). Columns and panels as described in Fig 2B–2C. Lags occurred below the *'mutational boundary'*, whereas the innovation of flight and scaling of metabolic rate reduced their magnitude. Below the mutational boundary, miniaturised lineages had longer lifespan and lower probability of dying from cancer than expected for their size, life history, and metabolism. Parameter values are as reported in Fig 2.

Finally, just above the mutational boundary, we also observe runs in which only very few substitutions got fixed, as discussed above. In many of these cases, cancer defences evolved to be lower than optimal (e.g. Fig 2F, 2G), such that lifespan evolved to be shorter (red parts near the mutational boundary in Fig 3B) and fitness evolved to be higher (Fig 3A) than in the 'complete lag' case that occurred when there was no substitutions. Lower than optimal cancer defences can be more adaptive than excessive defences, if the cost-benefit balance shifts to a net benefit because of higher reproductive success of cancer-prone organisms. On the other hand, if a subsequent need to increase the defences emerges, e.g. due to flight, a low mutation rate can hinder innovations to increase the previously-decayed cancer suppression and therefore, can, once more, produce fitness lags (see below). As such, intermediate mutation rates can lead some lineages to have an evolutionary mismatch in their cancer suppression at the end of miniaturisation and/or for extended periods of time during the miniaturisation process (as seen in Fig 2D–2G).

### Flight innovations and metabolic scaling reduce the fitness lags

Next, we explored whether flight, by reducing the extrinsic mortality rate in the shrinking lineage, can affect the evolution of cancer suppression. Flight indeed reduced the fitness lags: flying counteracted the otherwise shortening lifespan as body size reduced (longer lifespan in flighted lineages than flightless ones in Fig 3D), and kept the ancestrally high cancer defences, which delayed the onset of cancer to late ages, useful (no or lower fitness lags in flighted than flightless lineages in Fig 3A). The macroevolutionary timing of flight acquisition did not affect the results (Fig E in S1 Text). A smaller $r$, implying a greater reduction in extrinsic mortality rate upon flight acquisition, lowered fitness lags further, but did not fully compensate for the defence lags (Fig E in S1 Text), when we used values in line with those found in the literature [35,37].

Retaining ancestral defences is one of two ways birds can exhibit evolutionary lags in their cancer defences. The alternative scenario involves birds having already evolved lower cancer defences (suited to a small mammal-like life history, i.e. small-bodied and flightless) before the evolution of flight. This scenario would, for extant birds, predict lower than optimal cancer suppression, but it may be as a whole unlikely because it would combine fast enough evolution (to lower defences) in the pre-flight era with constraints acting on elevating defences again post-flight. However, given that mutations are more likely to destroy than to elevate existing defences, this remains a possibility, and in our model it is particularly relevant near the mutational boundary, where the evolution of lower than optimal cancer defences was likely (Fig 2D–2G).

We also explored how scaling of metabolic rate with body size affects cancer defence evolution, assuming metabolic rate correlates with the rate of oncogenesis. Lineages without metabolic scaling had overall the lowest probability of dying from cancer compared to all the lineages (constant $k$ in Fig 3C), including the small mammal-like lineage (Fig G in S1 Text), but also the largest evolutionary lags (Fig 3A). Metabolic scaling makes ancestral cancer defences more favourable than in the case without it, but does not fully diminish evolutionary lags (Fig 3A). However, the combination of metabolic scaling and flight results in a dinosaur-like life history in terms of cancer risk and lifespan (Fig 3C–3D; Fig G in S1 Text). As a result, the lineages where metabolic scaling and flight are combined have selection for dinosaur-level cancer suppression (Fig 2A) and virtually no evolutionary lags (lack of red in 'flighted, scaling $k$' in Fig 3A, compared to the other smaller panels).

Our results above suggest that innovations that reduce extrinsic mortality allow organisms to live longer and thus diminish the negative fitness effects of ancestrally high cancer suppression. If this is combined with an increasing cancer risk due to metabolic scaling, ancestral defences can be adequate for extant birds' life history. In this case, birds could have 1) kept dinosaurian cancer defences with no fitness lags, 2) suppressed dinosaurian cancer defences until the evolution of flight and then re-activated them, 3) decayed ancestral cancer defences until the evolution flight and gained novel ways of cancer suppression compared to their dinosaurian ancestors afterwards, 4) lost cancer defences and have been unable to elevate them to the optimal levels.

## Discussion

Our model outlines the conditions required for a lineage's evolutionary past to be visible in the bodies of small organisms, in the context of a legacy of high cancer suppression and consequent long lifespans of birds. Given that defending against rarely occurring conditions may incur suboptimally high costs (a trade-off between probabilistic lifespan shortening and the deterministic costs paid to express the defence regardless of the outcome), extant birds may

have adaptively eroded their ancestrally high dinosaurian defences. Our model shows that this happens under a high mutation rate combined with a sufficiently strong trade-off. On the other hand, at lower rates that make evolutionary change mutation-limited, cancer-related adaptations may survive the miniaturisation process. It therefore remains a real possibility that the longevity and cancer-resistance observed in birds [48,49,52] relative to mammals reflect birds' dinosaurian ancestry.

Our analysis also points at complexities in the above argument, as we also consider the changes in selection that may happen as a result of the innovation of flight. Today, most small-bodied birds are volant (able to fly), which may prolong lifespan, yet this effect was absent before bodies were small enough to create values of wing loading (weight divided by wing area) that permit flight. From flight onwards, the retention of ancestral cancer defences is again more favourable than it was before, creating nonlinearities in the expected evolutionary lags. It is relevant that reducing a cancer defence (the pre-flight but small-bodied scenario) may be easier than regaining it post-flight, and if so, we expect extant birds not to be more cancer-resistant than mammals of equivalent body size.

There are at least two scenarios where we predict birds to have unique cancer defences not shared by mammals: at sufficiently high mutation rates, birds, once flighted, could have gained their own cancer-related innovations, alternatively, mutations eroding defences are sufficiently rare that extant birds keep Mesozoic theropod adaptations. While comparative oncology has made progress in understanding avian cancer (e.g. [50] showing support for our premise that high reproductive effort trades off with cancer defences, discussed below), the search for unique adaptations has not been a priority; rather, researchers have focused on homologous traits or genes (shared between birds and mammals), either because of medical interest [74,75] or because, despite an interest in bird-specific questions, the known role of a gene in cancer comes from mammals, the better studied taxon [76].

Our model joins a tradition of others that rely on a trade-off between current reproductive success and the ability of an organism to delay the onset of somatic decay [60,77], generalizable as a manifestation of antagonistic pleiotropy of ageing [43]. As such, perhaps counterintuitively at first sight, having evolved or retained better cancer defences does not always translate into higher fitness. Should there be a trade-off between cancer defences and reproductive success ([50] provide evidence there is), it is possible to be 'excessively cancer-robust', and reproduce too sluggishly, if the trade-off is strong enough. Whereas the strength of this trade-off in the context of cancer likely varies between taxa and remains to be better understood [77], recent studies show that cancer prevalence increases with litter size in mammals [14] and clutch size in birds [50], supporting the trade-off approach. More generally, trade-offs between longevity and reproductive success are common in nature (see [78] for a review, and [79] for a study showing that reproductive success increases telomere attrition rate in birds). Cancer is merely one way for a body to fail over time, and our model, being suitable to be modified for different failures resulting from insufficient somatic maintenance [80,81], could also shed light on other aspects of senescence.

For completeness, we included a potential increase in the rate of oncogenesis due to metabolic scaling in a model variant, although the relationship between metabolic rate and cancer risk remains unclear (e.g. no evidence that the lower mass-specific metabolic rate of larger dog breeds protects them against cancer; [82]). The allometric scaling of mass-specific metabolic rate has been hypothesized to contribute to cancer risk management of larger animals [22,83], since larger animals have higher energy expenditure overall but lower per gram of body mass [69,84]. Metabolic rate is also elevated for flighted birds (compared with flightless ones and mammals; [85]), with estimates for the timing of increase roughly coinciding with the evolution of flight [86]. In line with [62], we showed that metabolic rate alone is

insufficient to compensate for the variation in cancer risk that is caused by changing body sizes. However, if we do assume a positive relationship between cancer risk and metabolic rate, birds' dinosaurian ancestry may have enabled them to evolve a high metabolic rate without cancer risk posing a constraint, assuming that some of their ancestral cancer defences still remained intact.

Evolutionary lags (or examples of maladaptation) occur in nature in numerous systems [87]. They are often studied in settings where the interests of two entities do not align, preventing (at least) one party from reaching an optimal situation at any time, (e.g. cuckoo-host arms races; [88]), or where environmental changes are so fast that they result in mismatches to the current environment (e.g. high cancer risk in post-industrial humans; [59,77]). We have expanded this view to macroevolutionary time scales, necessitating us to consider very large mutational ranges to reveal when evolution is mutation-limited or not. Mutation rates as a whole are much higher than the lowest range we consider [89], but only a small subset of all mutations will play a role in traits that modulate cancer risk, and improved defences are yet another subset of those. The exact placement of the mutational boundary will depend on model assumptions (e.g. costliness of cancer suppression, the area used to derive the effective population size from population density, or pleiotropic effects).

Here, the pleiotropic assumption that we used — a straightforward trade-off between unidimensional cancer defence level that reduces the rate at which oncogenic steps are completed and an equally unidimensional reproductive success — is a rather stark simplification of real cancer suppression which requires a complex network of genes (e.g. [90]). Intriguingly, some of the pleiotropic effects of these genes could effectively constrain the decay of cancer defences, illustrated by the naked mole rat example. These organisms have burrow-living adaptations, which seem to have been co-opted as cancer defences [25,29]. One can therefore speculate that decay in these defences would be selected against as long as they continue to live underground, even if the optimal level of cancer suppression in this species decreases. It is similarly conceivable that birds' dinosaurian ancestors had incorporated a trait evolved for a different function into their cancer defence network, and losing that trait in the bird lineage might be challenging if this function still exists and/or primed the evolution of further innovations [91]. For instance, speculatively, if a dinosaurian cancer suppression trait has been incorporated to manage energetic demands related to flight in birds, there will be selection against decays in this trait in birds, as long as they fly and do not evolve a novel evolutionary adaptation that replaces this trait, even if keeping higher than optimal cancer defence results in lower than maximal reproductive success. In line with this idea, several life-history traits, which also affect reproductive success, seem to adapt slowly to changing conditions in passerines (i.e. they show phylogenetic inertia; [92]) or birds in general [93].

Here, we studied the evolution of cancer suppression in birds during their miniaturisation from relatively larger-bodied ancestors. Depending on the costliness and mutation rate of cancer defences, as well as different scenarios in terms of flight and metabolic scaling, there are multiple trajectories that the cancer defence evolution could have followed. However, given the lower cancer incidence in birds compared to mammals and reptiles [16,48,49], birds having lower than optimal cancer suppression or no lags in cancer defences, either because of having evolved to the optimal level or due to metabolic scaling and flight, seems less plausible. In either case, bird-mammal contrast offers an exciting opportunity to discover hitherto unknown cancer suppression mechanisms, regardless of whether these have been maintained from dinosaurian ancestors, or represent their own unique innovations like those in the naked mole rats. In this study, our goal was to introduce an innovative macroevolutionary model that integrated different concurrent processes and to show how precise outcomes of different scenarios can be quantified. With the growing availability of genomic, demographic,

and oncological data across species, comparative analyses between birds and against other vertebrates are now very well-positioned to uncover the mechanisms underlying avian cancer robustness. In turn, complementing these comparative analyses with modelling approaches like ours can reveal other avenues of future research for novel cancer defence mechanisms beyond those in birds.

## Supporting information

**S1 Text. Robustness analysis of our results to different model assumptions.**
(PDF)

**S1 Data. The node dates and associated body sizes.** Extracted from [54] as described in the text. The values used in the simulations are shown in red.
(XLSX)

## Acknowledgments

The authors thank Athena Aktipis, Carlo C. Maley, and the other members of the Arizona Cancer Evolution Center, as well as the Kokkonuts group, for lively discussions.

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
