## [Decision Letter · Decision Letter 0]

23 Aug 2024

Dear Dr. Erten,

Thank you very much for submitting your manuscript "Bird size with dinosaur-level cancer defences: can evolutionary lags during miniaturisation explain cancer robustness in birds?" for consideration at PLOS Computational Biology.

As with all papers reviewed by the journal, your manuscript was reviewed by members of the editorial board and by several independent reviewers. In light of the reviews (below this email), we would like to invite the resubmission of a significantly-revised version that takes into account the reviewers' comments.

We cannot make any decision about publication until we have seen the revised manuscript and your response to the reviewers' comments. Your revised manuscript is also likely to be sent to reviewers for further evaluation.

Sincerely,

Andrea Ciliberto

Academic Editor

PLOS Computational Biology

Tobias Bollenbach

Section Editor

PLOS Computational Biology

Reviewer's Responses to Questions

**Comments to the Authors:**

Reviewer #1: This work models multiple possible scenarios for the evolution of cancer defences in birds after divergence from their dinosaur ancestors.

A series of macroevolutionary simulations are run to recreate multiple possible scenarios to investigate the potential impacts of body size reduction and the gain of flight capability may have on evolution of cancer defences in birds.

Many of the parameters used are carefully defined and justified by the authors based on the literature and evolutionary principles.

However, there was little information provided regarding the basis on which the range of values for m (mutation rate per genome per generation) were selected.

This is important because the authors show there is a clear ‘mutation rate boundary’ effect that leads to dramatically different outcomes depending on whether rates of mutations that impact cancer defences are high or low.

Likewise, it is unclear how many loci impact cancer defences in these simulations, and what their combined length is.

While anti-cancer mechanisms are complex and the total number of possible loci controlling them is likely not known at this point, the number of length of canonical tumour suppressor genes would serve as a useful baseline, given the strong conservation of their sequence and function across vertebrates.

The simulations yielded intriguing and insightful findings that provide understanding of the evolutionary tradeoffs between cancer defence and other advantageous traits in birds.

However, overall the paper is somewhat inconclusive in regards to what is the most likely scenario of evolution of cancer defences in birds. Other than ruling out the loss and subsequent regain of cancer defences as not being parsimonious, it’s hard to tell which evolutionary scenario modelled here is most likely.

The manuscript would be improved by data that could support and solidify its main messages.

Numerous bird genomes are now available, covering a range of lineages. These data could be used for comparative analysis among birds and against other vertebrates classes, particularly in the aforementioned canonical tumour suppressor genes, to determine whether mutation rates correspond to capability for flight and body size.

Beyond that, data on lifespan, brood size and cancer incidence could be compared within both flighted and flightless bird lineages that show large variation in body size, to assess validity of other associations predicted by the authors’ models.

Reviewer #2: This manuscript by Erten et al. explores the evolutionary dynamics of cancer suppression in birds, proposing that birds may have retained ancestral cancer defenses from their large-bodied, dinosaurian ancestors. The study models how innovations like flight and metabolic scaling affect cancer risk and longevity, suggesting that these adaptations helped maintain or re-activate high cancer suppression in birds, despite miniaturization. The findings provide insights into the relationship between evolutionary history, life history traits, and cancer resistance in birds. The manuscript is well written, logical and I only have a few, rather minor comments.

Line 5 (also abstract): While it was long believed that cell division is one of the key sources of mutations, it seems that most-mitotic cells also accumulate mutations at pretty much similar rates as dividing ones. Read this: https://www.nature.com/articles/s41586-021-03477-4

Line 13: a priori should be italicized

Line 31: birds experience not just a delayed onset of senescence, but also a slower rate of senescence: see e.g. https://royalsocietypublishing.org/doi/full/10.1098/rstb.2019.0744

Line34: "experience lowered extrinsic mortality from predation" - not just predation. Flying and migration over long distances serves to avoid food shortages, so also , a pattern repeated in 34 flying mammals [32,34,37] which, too, live long for their body size.

Line 497-500: long and difficult to follow. e.g. " incorporated burrow-living adaptations into their cancer defence" - burrow-living is not a cancer defence I believe, or at least it's not the primary reason. Also " losing the defence would be selected against" could be said as "maintinting the defence would be selected for" to simplify the text.

Line 501.504: an example could help to understand what you refer here.

The manuscript argues that metabolic rate in birds increased compared to their dinosaur ancestors. Are the authors sure of this? I believe metabolic rate closely correlates with body mass across all vertebrates. do the authors maybe mean mass-specfific metabolic rate? Which is indeed higher in birds than in size matched mammals/reptiles, etc. If so, all text and figures need to be correled accordingly.

Reviewer #3: Review of 'Bird size with dinosaur-level cance defences: can evolutionary lags during miniaturisation explain cancer robustness in birds' from Erten et al for PLoS Comp Biol

The authors ask whether the reduced frequency of cancer in existing birds may be a heritage of dinosaurs, which may have had higher defences against cancer due to their large size. To address this possibility, they develop a mathematical model to produce macroevolutionary simulations. The hypothesis being correct is not trivial. Higher cancer defences come with reproductive costs, which will be typically counter selected. Using the model, the authors identify scenarios where this is actually possible: if costs are minimal (obviously); if birds undergo additional changes (here, metabolic) that increase the chances of developing cancer; and if they acquire additional properties (here, the ability to fly) that may decrease the extrinsic death rate and allow a longer life-span.

The paper is interesting. Although the authors took commendable efforts to identify reasonable parameter ranges, the results are very speculative. Yet, I believe this is appropriate for PLoS Computational Biology. The manuscript has the merit to discuss an intriguing idea, and thus I support it for publication. Yet, I think it can be improved, especially in terms of data representation.

Major comments

1 - The model has a large number of parameters. The authors decided to investigate the model changing two of them: mutation rate and cost of cancer defence. What is the rationale for choosing these two?

2- After reading the paper, the answer to the question very clearly stated in the title is not super clear, at least to me. One is left with the impression that almost everything is possible. A few suggestions/questions:

- is there any evidence that mutation rate and cost can vary in the range they use for the density plots? Which of the many scenarios is closer to available data? And so, which area of the density plots is more realistic?

- the abstract could be more direct making explicit the 'under certain assumptions' cited there.

3- One general point: changes in metabolism may increase the chances of developing cancer, and thus, the authors argue, would avoid eroding cancer defences in shrinking lineages. In this case, would one expect a different or similar tendency to develop cancer in birds and in mammals? More in general, if miniaturisation, flight, scaling k represents existing birds and constant small, flightless constant k is mammal-like, in which parameter range do the first show less predisposition for cancer? From Figure S6 one would not notice big differences for DL and DW. Figure 4B and 2d also do not show big changes between these two conditions. I think this point (comparison mammal-like vs birds in terms of cancer propensity in the model) should be better discussed, since the original observation is precisely the difference in cancer propensity between birds and equally sized mammals.

4- I do not fully get the paradox about life span and cancer defence. Apparently Figure 4a shows both behaviors (p 10 "even as they continued to prolong life span (Fig 4a)" and "typically do not live long enough to enjoy...(Fig 4a)). This should be better explained (see later on data visualization). Trying to understand, I looked for the equations, and unless I am mistaken I could not find the explicit form of L(d) (only a reference to [54]). Also, in p7 L(d) is called 'reproductively active lifespan', and elsewhere simply as lifespan. This is an important point, since if I understand the two can be different (e.g., in introduction: longer life span, lower cancer risk but reduced reproductive success). Please, clarify.

5 - I find the data visualization less than optimal. All data are in few Figures packed with density plots, and refereed generically in the main text as, eg, Figure 3 or Figure 3a, which contain lots of subplots. I had to go through them several times before getting the points. Hereafter, I give some criticisms and suggestions on how to improve them:

-p 10, different references to the same Figure 3C point to quite different things: "being constrained to express ancestral defences did not incur large fitness losses for the miniaturised lineages"; and "maintaining ancestral defences became maladaptive". This is confusing also because Figure 3C is made of four different density plots. In Figure 2 and 4 that are 6. Which of these plots are being referred to? Within the relevant density plots, I think it would help the reader if the three main quadrants were identified (low costs; high costs with yes/no substitutions; and so concerning my question, adding in the lower right corner 'maladaptive' and other labels in the other two regions).

- In figure 3 we have 4 density plots (two missing are in S6), in Figures 3 and 4 we have six of them. Why is that?

- The most interesting comparisons are pairwise. If the Authors want to keep all density plots together, maybe they could organize them as a 2X2 table where pairwise comparisons are

more easily detected

(something like

scaling k constant k

flight

flightless

).

I would then keep the two controls (ie, constant [small or large] constant k flightless) outside the table. Also, the meaning of the controls (eg, the mammal-like) may be stated explicitly in the methods section where they are introduced (p4).

- Figure 2b-c, some of the low mutation rate trajectories are not easy to follow (do they go up and down?).

- p 10 "all lineages evolved to their optimal level of cancer defences (Fig 2e)". Where do we see the optimal level in 2e?

- p 11 "kept the ancestrally high cancer defences useful Fig3", not clear to me where I see this in Fig 3.

Minor comments:

- p3 "The life history option of a slow life history" needs to be rephrased

- p4

- "we contrast results" -> " we compare results"

- end of introduction "lowering the effects of these evolutionary lags" should not it rather be 'diminuishing the evolutionary lags'?

- in the text on [46] -> in the text [46]

- "size and date data" -> not clear

p6

- the concept of "rate of oncogenic steps" should be better introduced and given appropriate citations (eg, Vogelstein).

- equation (1) is important, and derived in a previous paper. It should be briefly explained in this text to make it 'self-sustained'.

- the fact that flightless, small sized from time 0 and constant k is a mammal-like lineage appears all of a sudden at p.9. The concept is relevant, and it should be introduced earlier (see main comment 5).

- If I am not mistaken, Figure 3b is never mentioned in the text. We find a description of its meaning in the legend. I think this part belongs to the text. In general, captions bear information that could be moved to the main text.

- References to supplementary material should indicate precisely which figure in there they refer to.

- The first paragraph of Results is a new abstract, quite difficult to follow.

- Legend in Figure 2a, flighted -> flight, to be consistent with all other figures

**Have the authors made all data and (if applicable) computational code underlying the findings in their manuscript fully available?**

Reviewer #1: Yes

Reviewer #2: Yes

Reviewer #3: Yes

PLOS authors have the option to publish the peer review history of their article (what does this mean?). If published, this will include your full peer review and any attached files.

Reviewer #1: **Yes: **Dr David Goode

Reviewer #2: No

Reviewer #3: No
---

## [Decision Letter · Decision Letter 1]

27 Mar 2025

PCOMPBIOL-D-24-01081R1

Shrinking to bird size with dinosaur-level cancer defences: evolution of cancer suppression over macroevolutionary time

PLOS Computational Biology

Dear Dr. Erten,

Thank you for submitting your manuscript to PLOS Computational Biology. After careful consideration, we feel that it has merit but does not fully meet PLOS Computational Biology's publication criteria as it currently stands. Therefore, we invite you to submit a revised version of the manuscript that addresses the points raised during the review process.  In particular, we ask you to address the reviewers' concern about the readability of the manuscript, both for concerning the text and the figures as indicated by reviewer 3; and to include more exensively in the revised manuscript the arguments discussed in the answer to reviewer 1.

Please submit your revised manuscript within 30 days May 27 2025 11:59PM. If you will need more time than this to complete your revisions, please reply to this message or contact the journal office at ploscompbiol@plos.org. Please include the following items when submitting your revised manuscript:

We look forward to receiving your revised manuscript.

Kind regards,

Andrea Ciliberto

Academic Editor

PLOS Computational Biology

Tobias Bollenbach

Section Editor

PLOS Computational Biology

**Reviewers' comments:**

Reviewer's Responses to Questions

Reviewer #1: I appreciate the authors' detailed and thoughtful responses to my comments & concerns. They have enhanced and clarified my understanding of the design and intent of their work.

However, I would have liked the information in their responses to be incorporated directly into the revised manuscript in some way, beyond the addition at lines 606-609 regarding potential pathways for future use of their model to inform genomic and phenotypic data.

I feel this would improve the ability of first-time readers to understand and appreciate the intent and approach of this study, and thus be more likely incorporate the authors' frameworks and ideas into their own research.

The 2024 paper by Kapsetaki et al actually does provide some validation of the sort I proposed. It was not mentioned in their response to my comments but I see was referred to in response to other reviewers and is referenced in the manuscript.

The association between brood size and cancer incidence across birds is encouraging. But Kapsetaki did not report any association of cancer rates with body size, which was one of the key elements of the authors' model. I'd therefore be interested in the authors' perspective on how that negative findings aligns with their models predictions, which I feel was further enrich this paper.

Reviewer #3: The revised version of the manuscript in my opinion is only relatively improved.

As before, I find the paper interesting, but it is hard to read and to follow.

All the suggestions given for improving the figures have not been accepted, which

can be fine of course, provided the Authors had tried alternative ways to improve the paper.

This, it seems to me, was not done.

1- I understand that the model is complex, it includes many features and it is hard to explain.

Exactly for this reason it would make sense to introduce the reader to the questions more

clearly. The first paragraph of the results 'Reduced optimal cancer suppression in shrinking lineages' is very hard

to follow. There is no introduction about the question addressed. Figure 2a requires a knowledge about

almost everything: substitutions, lifespan, metabolic scaling, flight...to further complicate things, the reader is

taken to Figure 4b before any other Figure (the generic reference Figures 2-4 cover the whole paper).

I find it much easier to start the paper from the second paragraph. Many of the concepts

introduced in the first paragraph are introduced later again anyway. I am not suggesting to remove the

first paragraph, but I invite the authors to be more pedantic in what they ask and what they find, assuming that some readers may well

decide to start reading from the Results section. A starting point is to make sure the Figures are cited

in the correct order in the text.

2 - I still think that it would be very useful to have the figures in panels 2X2

scaling constant

flightless panel panel

flight panel panel

so that on the x on the scaling changes and on the y only the ability to fly. For example,

it would emerge very clearly that flight has the largest effect on lifespan. The two additional

panels with large vs small can be added in the same figure.

Other comments:

1 - In the text again and again it is reported that increased metabolic rate goes together

with increased risk of cancer (from the abstract onward: "an increase in cancer risk due to metabolic scaling

"). Then we find in the Discussion and in the rebuttal that the model holds also when this is not the case

"our results remain qualitatively similar whether metabolic 485 rate is assumed to correlate with cancer risk or not". I find this confusing.

The authors should clarify this point.

2- the authors write about 'realistic' mutation rates (line. 95) and about 'more realistic scenario' (line 355), but in their rebuttal they argue, if I understand from their answers, that giving realistic ranges is not possible.

3- line 320: what does 'importance' mean? Where do I see this?

4- I am not sure what the Authors mean for pleiotropy, it is used to explain the mutation limited regime, but also in alternative to it (line 87).

5- line 309: cancer risk decreased -> compared to?

6- Figure 2b and 2c are not changed, and remain hard to understand to me. I do not get the blu

trajectories: do they go up and down? Only down? Is it possible to find a representation that shows

clearly the dynamics of each trajectory?

7- The last raw of Figure 4 was not introduced in the text, unless I am mistaked.

8- I find the Discussion very long, when compared to the Results section.

**Have the authors made all data and (if applicable) computational code underlying the findings in their manuscript fully available?**

Reviewer #1: Yes

Reviewer #3: Yes

PLOS authors have the option to publish the peer review history of their article (what does this mean?). If published, this will include your full peer review and any attached files.

Reviewer #1: **Yes: **Dr David Goode

Reviewer #3: No

**Figure resubmission:**
---

## [Decision Letter · Decision Letter 2]

13 Aug 2025

Dear Dr. Erten,

We are pleased to inform you that your manuscript 'Shrinking to bird size with dinosaur-level cancer defences: evolution of cancer suppression over macroevolutionary time' has been provisionally accepted for publication in PLOS Computational Biology. We only ask you to corrcet a typo. At line 406, there is a reference to Figure 3d,f,g

which is incorrect, since there is neither Figures 3f nor 3g. Please, correct it in the final version of the manuscript.

Before your manuscript can be formally accepted you also will need to complete some formatting changes, which you will receive in a follow up email. A member of our team will be in touch with a set of requests.

Best regards,

Andrea Ciliberto

Academic Editor

PLOS Computational Biology

Tobias Bollenbach

Section Editor

PLOS Computational Biology

Reviewer's Responses to Questions

**Comments to the Authors:**

Reviewer #3: The Authors have done a good job in improving the readability of the manuscript.

I think the flux of the arguments is now much improved, and I found the figures

more readable. I think the paper is very interesting and ready for being published.

There is just one last typo: line 406, they refer to Fig 3d,f,g

but there are no figures 3f and 3g. This needs to be corrected.

**Have the authors made all data and (if applicable) computational code underlying the findings in their manuscript fully available?**

Reviewer #3: Yes

PLOS authors have the option to publish the peer review history of their article (what does this mean?). If published, this will include your full peer review and any attached files.

Reviewer #3: No

---

## [Editor Report · Acceptance letter]

PCOMPBIOL-D-24-01081R2

Shrinking to bird size with dinosaur-level cancer defences: evolution of cancer suppression over macroevolutionary time

Dear Dr Erten,

I am pleased to inform you that your manuscript has been formally accepted for publication in PLOS Computational Biology. Your manuscript is now with our production department and you will be notified of the publication date in due course.

With kind regards,

Anita Estes
